# Determinants of Tunneled Hemodialysis Catheter Implantation Time by Ultrasound Guidance: A Single-Center Cross-Sectional Study

**DOI:** 10.3390/jcm11123526

**Published:** 2022-06-19

**Authors:** Désirée Tampe, Björn Tampe

**Affiliations:** Department of Nephrology and Rheumatology, University Medical Center Göttingen, 37075 Göttingen, Germany; desiree.tampe@med.uni-goettingen.de

**Keywords:** retrograde-tunneled hemodialysis catheter, antegrade-tunneled hemodialysis catheter, rapid atrial swirl sign, ultrasound-guided tip positioning, vascular access, end-stage kidney disease, kidney replacement therapy

## Abstract

Background: We have previously reported that the ultrasound (US)-guided tip positioning is an accurate and safe procedure for placement of retrograde- and antegrade-tunneled hemodialysis catheters (HDCs). However, determinants of tunneled hemodialysis catheter implantation time by using US guidance have not been described yet. Therefore, we here report a comparative analysis to identify determinants of implantation time for retrograde- and antegrade-tunneled HDCs placement by US guidance. Methods: We performed a cross-sectional study to compare implantation time for US-guided tip positioning of retrograde- and antegrade-tunneled HDCs. We included a total number of 47 tunneled HDC insertions, including 23 retrograde tunneled and 24 antegrade-tunneled HDCs in patients requiring placement of an HDC for the temporary or permanent treatment of end-stage kidney disease (ESKD) in a single-center, cross-sectional pilot study. Results: We show that clinical and laboratory parameters did not differ between retrograde- and antegrade-tunneled HDC implantations. There was a tendency for shorter implantation time in antegrade-tunneled HDCs, although not statistically significant. Finally, we identified an independent inverse association between body weight (BW) and platelet counts with HDC implantation time specifically in antegrade-tunneled HDCs. Conclusion: In this study, we identified determinants for tunneled HDC implantation time that might be relevant for patients and interventionists.

## 1. Introduction

In patients with end-stage kidney disease (ESKD), vascular access is required for kidney replacement therapy (KRT) [1]. When arteriovenous access is not available, tunneled hemodialysis catheters (HDCs) are the preferred vascular access in patients requiring KRT for more than two weeks or those who develop ESKD based on a lower risk for infectious complications compared to non-tunneled HDCs [2]. Traditionally, fluoroscopic guidance is used to ensure the correct placement and positioning of the tunneled HDC [2]. Live fluoroscopic guidance allows for real-time positioning, minor adjustments, and safe placement of tunneled HDCs [3]. However, this method imposes additional safety risks to the patient and operator due to the radiation [4]. It has already been shown that the conversion from a non-tunneled to a tunneled HDC without using fluoroscopy in an incident dialyzed cohort is safe, but also inexpensive [5]. We have previously reported that the ultrasound (US)-guided tip positioning is an accurate and safe procedure for placement of retrograde- and antegrade-tunneled HDCs [6,7]. Previous observations reported that the duration of peripheral and pulmonary arterial catheter placement time was a risk factor for the development of infections, independent of the neutropenic status, or the administration of antibiotics during catheterization [8]. In the current study, we aimed to identify determinants of implantation time for retrograde- and antegrade-tunneled HDCs placement by US guidance.

## 2. Materials and Methods

### 2.1. Study Population and Setting

We included a total number of 48 patients from January 2020 to April 2022 who required placement of a tunneled HDC for the temporary or permanent treatment of ESKD admitted to our Department of Nephrology and Rheumatology at the University Medical Center Göttingen, Germany (protocol number 3/6/21). Informed written consent was obtained from all subjects involved in the study for use of routinely collected data for research purposes as part of their regular medical care in the contract of the University Medical Center Göttingen, Germany.

### 2.2. Catheter Placement Procedure and Material

Due to the manufacturer’s reasons, retrograde-tunneled HDCs were not available and antegrade-tunneled HDCs were alternatively implanted between June 2021 and April 2022. For the placement of the retrograde-tunneled HDC, Palindrome™ Precision RT-reverse tunneled catheters (Medtronic, Minneapolis, MN, USA) were used. We used 15 french (F)-sized HDCs that were 23, 28, or 33 cm in length from tip to cuff, depending on the patient’s height and the site of insertion (right or left). For the placement of the antegrade-tunneled HDC, Palindrome™ Precision Symmetric Tip Dialysis Catheters (Medtronic, Minneapolis, MN, USA) were used. We used 14.5 F-sized HDCs that were 23 or 28 cm in length from tip to cuff, depending on the patient’s height and on the site of insertion (right or left). After obtaining informed consent from the patient, the procedure was performed by two interventionists with continuous hemodynamic monitoring in a dedicated area of our ICU to ensure maximum sterility and patient safety. All HDC implantations were performed by a single interventionist (B.T.) with assistance. The right internal jugular vein (IJV) was the preferred access site. After sterile preparation and draping, local anesthesia with 2% mepivacaine hydrochloride was applied and the IJV puncture was performed under US guidance (GE Venue US machine, General Electric Company, Boston, MA, USA) using a sterile probe cover with an out-of-plane approach. After venous cannulation, a guide wire was inserted for venous dilation and the HDC was inserted through the peel-apart introducer sheath. After tip positioning using RASS and exit site definition, an exit site incision was performed and the HDC was tunneled retrograde or antegrade under local anesthesia and inserted after peeling away the introducer sheath. After placement of the tunneled HDC, conventional anterior chest radiography was performed to document the correct placement of the catheter tip.

### 2.3. Ultrasound Visualization and RASS

Focused B-mode echocardiography using the subcostal (SC) view was used to visualize the right atrium (RA) and right ventricle (RV), and echocardiography was performed using a sector probe of a GE Venue US machine (General Electric Company, Boston, MA, USA). After HDC insertion, a 10 mL normal saline flush was injected by one of the interventionists, while echocardiography was performed by a third operator skilled in echographic examinations. The appearance of the saline swirl entering the RA within one second of the start of the saline flush was interpreted as correct HDC tip positioning, as previously reported [6,7,9].

### 2.4. Assessment of HDC Implantation Time

The total HDC implantation time was defined between the first venous puncture and final catheter placement. The procedural HDC implantation time was defined between venous guide wire insertion and final catheter placement.

### 2.5. Patient Consent and Ethics Approval

The study included patients aged >18 years of age, all patients provided written informed consent for all procedures presented in this paper. The study was conducted according to the guidelines of the Declaration of Helsinki and approved by the Ethics Committee of University Medical Center Göttingen (protocol number 3/6/21, approval date 25 June 2021).

### 2.6. Statistical Analysis

Descriptive statistics with frequencies and percentages were used for the characterization of the study cohort. Continuous variables are expressed as median and IQR, categorical variables are presented as frequency and percentage. For group comparisons, the Mann–Whitney U-test was used to determine differences in medians. Non-parametric between-group comparisons were performed with Pearson’s Chi-square test. Spearman’s correlation was performed to assess correlations and heatmaps reflect the mean values of Spearman’s ρ. A Spearman’s ρ more than ±0.4 in the correlation matrix was defined as relevant indicated by rectangle boxes, and independent statistical evaluation of these parameters was performed by linear regression. A probability (*p*) value of <0.05 was considered statistically significant. Data analyses were performed with GraphPad Prism (version 9.3.1 for MacOS, GraphPad Software, San Diego, CA, USA), and linear regression analyses were performed using IBM SPSS Statistics (version 27 for MacOS, IBM Corporation, Armonk, NY, USA).

## 3. Results

### 3.1. Study Population

We included a total number of 48 patients, one patient was excluded due to failure of HDC insertion (Figure 1). In the remaining 47 patients, 50 HDCs were implanted and three HDCs were excluded because of replacement after the first insertion due to HDC dislocation in the same patients (Figure 1). For the final analysis, 47 HDC insertions were included, separated into 23 retrograde-tunneled and 24 antegrade-tunneled HDCs (Figure 1).

Clinical and laboratory parameters including demographic data and etiology of ESKD in the cohort of the 47 included patients are included in Table 1. Among these parameters, demographic data were comparable between patients receiving a retrograde- and antegrade-tunneled HDC implantation. Furthermore, etiologies of ESKD were equally distributed in both groups. Blood coagulation parameters did not differ between retrograde- and antegrade-tunneled HDC implantations, only a minor fraction had thrombocytopenia at the time of HDC implantation not differing between the groups. In addition, there was a comparable distribution of vascular access sites, catheter lengths, and performance at first treatment after HDC implantation in both groups.

### 3.2. Comparison of HDC Implantation Time in Retrograde- and Antegrade-Tunneled HDCs

We next compared total and procedural implantations times between retrograde- and antegrade-tunneled HDCs. Although there was a tendency towards shorter total and particularly procedural implantation time for antegrade-tunneled HDCs with a median of 30.5 (25–40.75) minutes as compared to retrograde-tunneled HDCs with 46 (25–56) minutes, this difference was not statistically significant (Figure 2A,B).

### 3.3. Determinants of HDC Implantation Time in Retrograde- and Antegrade-Tunneled HDCs

We finally aimed to identify determinants of HDC implantation time in the total cohort and separated for retrograde- and antegrade-tunneled HDCs. Specifically for antegrade-tunneled HDCs, we observed an independent inverse association between body weight (BW) and platelet counts with HDC insertion time as confirmed by multiple linear regression analysis (Figure 3A–C and Table 2).

## 4. Discussion

Tunneled HDC implantation time by fluoroscopic guidance has previously been described at 29 min, implicating that HDC insertion by US guidance might require a longer time [10]. However, a direct comparison of these techniques regarding implantation time has not been described yet and requires further investigation. It has recently been shown that tunneled HDC insertion without fluoroscopy was safe and had comparable short-term outcomes as compared to fluoroscopy-guided HDC insertion [11]. Interestingly, catheter patency was longer in the non-fluoroscopy group, even after adjustment for history of previous catheter use and catheter access site [11]. Additionally, the mean length of hospital stay was reduced in the non-fluoroscopy group [11]. We here show that clinical and laboratory parameters did not differ between retrograde- and antegrade-tunneled HDC implantations. Our observation that antegrade-tunneled HDC implantation tends to be faster might be of relevance, and the identification of patient subgroups who might benefit from especially antegrade-tunneled HDCs requires further investigation. This is especially relevant since previous observations reported that the duration of peripheral and pulmonary arterial catheter placement time was a risk factor for the development of infections, independent of the neutropenic status, or the administration of antibiotics during catheterization [8]. Our observation that antegrade-tunneled HDC implantation tends to be faster needs further validation regarding the risk of the development of infectious complications. On the other hand, the retrograde-tunneled technique has several advantages over the antegrade-tunneled HDC insertion technique [12]. First, the HDC tip position is established first and therefore consistent. Second, the cuff never passes through the exit site and requires only a small incision that prevents bleeding and the accidental removal of the cuff before it is incorporated. Third, the hub is detachable, and the replacement of a damaged hub or clamps is easy to perform without disturbing a functioning catheter. Finally, we identified an inverse association between body weight and platelet counts with implantation time specifically in antegrade-tunneled HDCs. It has previously been shown that obesity significantly increased the operating times in carotid endarterectomy and central vascular procedures [13]. However, we did not observe a correlation with the body mass index (BMI) itself, and longer implantation time was associated with less body weight. Regarding platelet count, it has already been shown that preoperative platelet counts in idiopathic thrombocytopenic purpura impacts length of surgical procedures [14]. These observations require further investigation and confirmation in independent cohorts.

Our study has several important strengths. First, an important characteristic of our study design was to directly compare tunneled HDC implantation time by US-guided tip positioning separated for retrograde- and antegrade-tunneled HDCs. Second, a selection bias for either catheter can be excluded because retrograde-tunneled HDCs were not available due to manufacturer’s reasons. Third, all HDC implantations were performed by a single interventionist, excluding a procedural bias. Finally, relevant clinical and laboratory parameters were comparable in both groups. Our study has also several limitations. First, the relatively small number of patients in a single center and no randomization due to the unavailability of retrograde-tunneled HDCs require validation in independent prospective cohorts. Second, the lack of control groups for retrograde- and antegrade-tunneled HDCs using traditional procedures is an important limitation of the study. Third, all tunneled HDCs were inserted through the internal jugular veins, and different access sites (e.g., external jugular veins, subclavian veins, and femoral veins) might differ. Nevertheless, identifying determinants of tunneled HDC implantation time is of relevance and might contribute to better knowledge about patient subgroups who might benefit from either method. 

## 5. Conclusions

In this study, we identified determinants for tunneled HDC implantation time that might be relevant for patients and interventionists.

## Figures and Tables

**Figure 1 jcm-11-03526-f001:**
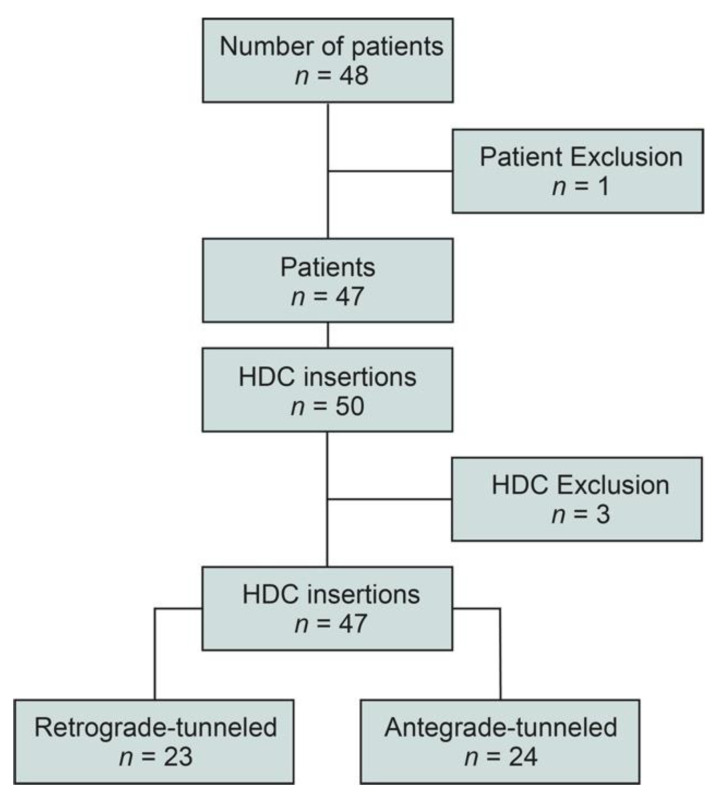
Total patient cohort. STROBE flow chart of patient disposition with the indication of tunneled HDC insertion. Abbreviations: HDC, hemodialysis catheter; STROBE, Strengthening the Reporting of Observational Studies in Epidemiology.

**Figure 2 jcm-11-03526-f002:**
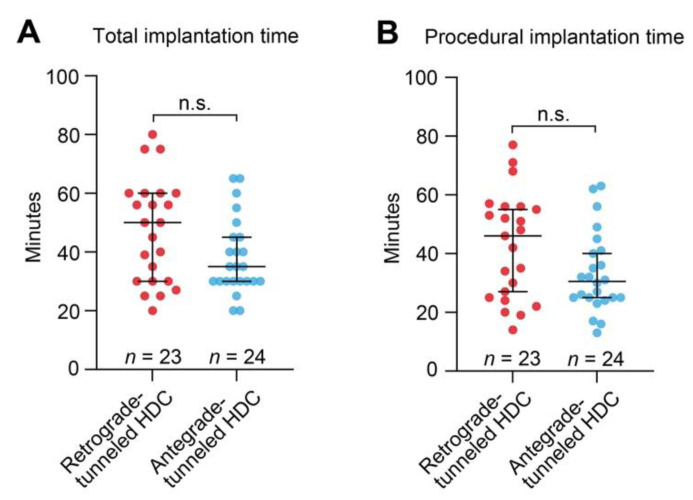
(**A**,**B**) Comparison of HDC implantation time in retrograde- and antegrade-tunneled HDCs. For group comparisons, the Mann–Whitney U-test was used to determine differences in medians. Abbreviations: HDC, hemodialysis catheter; n.s., not significant.

**Figure 3 jcm-11-03526-f003:**
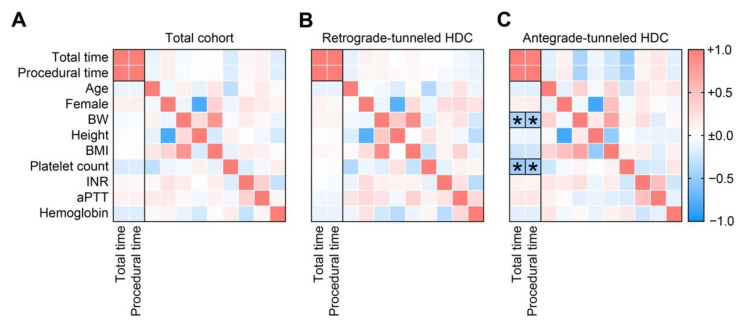
(**A**–**C**) Correlative analyses of HDC implantation time are shown by the heatmap reflecting mean values of Spearman’s ρ. The rectangle box indicates a Spearman’s ρ more than ±0.4, the asterisk a significant correlation in the multiple linear regression analysis (*p* < 0.05). Abbreviations: aPTT, activated partial thromboplastin time; BMI, body mass index; BW, body weight; cm, centimeter; HDC, hemodialysis catheter; INR, international normalized ratio.

**Table 1 jcm-11-03526-t001:** Clinical parameters in the total cohort of patients.

Demographic Data	Retrograde-Tunneled HDC	Antegrade-Tunneled HDC	*p*-Value
Median age (IQR)—years	71 (59–81)	65.5 (58.25–76.75)	0.6012
Female sex—no. (%)	7(30.4)	9 (37.5)	0.6094
Median height (IQR)—cm	172 (168–180)	173 (168–176.8)	0.9453
Median BW (IQR)—kg	85 (75.5–104)	81 (72.5–87)	0.1904
Median BMI (IQR)—kg/m^2^	29.41 (23.88–33.57)	27.27 (24.45–29.72)	0.2400
History of catheterization—no. (%)	1 (4.3)	2 (8.3)	0.5763
**Etiology of ESKD**			
Diabetic nephropathy—no. (%)	7 (30.4)	2 (8.3)	
Cardiorenal syndrome—no. (%)	4 (17.4)	7 (29.2)	
Hypertensive nephropathy—no. (%)	1 (4.3)	2 (16.7)	
Autoimmune disease—no. (%)	4 (17.4)	5 (20.8)	
Shock—no. (%)	4 (17.4)	8 (33.3)	
Others—no. (%)	3 (13)	0 (0)	0.1377
**Laboratory data**			
Platelet count (IQR)— ×1000/μL	188 (99–330)	177 (111.8–233.5)	0.5653
Thrombocytopenia—no. (%)	8 (34.8)	11 (45.8)	0.4403
INR (IQR)—ratio	1.2 (1–1.3)	1.2 (1.1–1.275)	0.7578
aPTT (IQR)—seconds	30 (28–36)	32 (29.25–35.5)	0.3256
Hemoglobin—g/dL	8.2 (7.7–9.5)	8.75 (7.725–9.625)	0.4694
**Vascular access**			
Right IJV—no. (%)	17 (73.9)	13 (54.2)	
Left IJV—no. (%)	6 (26.1)	11 (45.8)	0.1590
**Catheter length**			
23 cm—no. (%)	14 (60.9)	14 (58.3)	
28 cm—no. (%)	8 (34.8)	10 (41.7)	
33 cm—no. (%)	1 (4.3)	0 (0)	0.5484
**Catheter performance**			
Blood flow (IQR)—mL/min	200 (200–250)	250 (200–250)	0.1122

Continuous variables are expressed as median and IQR, categorical variables are presented as frequency and percentage. For group comparisons, the Mann–Whitney U-test was used to determine differences in medians. Non-parametric between-group comparisons were performed with Pearson’s Chi-square test. Abbreviations: aPTT, activated partial thromboplastin time; BMI, body mass index; BW, body weight; cm, centimeter; ESKD, end-stage kidney disease; HDC, hemodialysis catheter; IJV, internal jugular vein; INR, international normalized ratio; IQR, interquartile range; kg, kilograms; m, meter; no., number.

**Table 2 jcm-11-03526-t002:** Multiple linear regression of parameters associated with antegrade-tunneled HDC implantation time.

Parameters Associated with Total Implantation Time	SE	β	*p*-Value
BW—kg	0.2052	−0.3867	0.0311
Platelet count—×1000/μL	0.0196	−0.5447	0.0038
**Parameters Associated with Procedural Implantation Time**			
BW—kg	0.2239	−0.3758	0.0415
Platelet count—×1000/μL	0.0214	−0.5117	0.0075

Abbreviations: β, beta coefficient; BW, body weight; kg, kilograms; SE, standard error.

## Data Availability

Deidentified data are available on reasonable request from the corresponding author.

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
