# Peer review of "Determinants of Tunneled Hemodialysis Catheter Implantation Time by Ultrasound Guidance: A Single-Center Cross-Sectional Study"

_jcm, 2022, doi:10.3390/jcm11123526_

Round 1

Reviewer 1 Report

Désirée and Björn Tampe here present a single-center study comparing positioning of retrograde- and antegrade-tunneled hemodialysis catheters (HDC). There are few articles about this topic in literature and this kind of publication usually brings significant knowledges to the community. The procedure they used was previously evaluated for retrograde- (JCM | Free Full-Text | The Rapid Atrial Swirl Sign for Ultrasound-Guided Tip Positioning of Retrograde-Tunneled Hemodialysis Catheters: A Cross-Sectional Study from a Single Center | HTML (mdpi.com)) and then (The rapid atrial swirl sign for ultrasound-guided tip positioning of antegrade-tunneled hemodialysis catheters: A cross-sectional pilot study - Désirée Tampe, Marlene Plüß, Tim Kuczera, Björn Tampe, 2022 (sagepub.com)) for antegrade-tunneled HDC positioning in the same center, by the same authors. Now the authors compare the two types of HDC. They highlighted no differences between the two devices for implantation time. According to their observations, prolonged HDC insertion of antegrade-tunneled HDC was associated with decreased body weight and lower platelet counts.

First I have to alert the authors and the editor that this manuscript presents serious breach of scientific integrity. The authors published a study about retrograde-tunneled HDC in September 2021, and a study about antegrade-tunneled HDC in April 2022. Now they submit another study comparing the two devices. The ethics committee approval number was strictly the same. Hence, it really looks like Salami Slicing. Moreover, for only 12 references 2 of them are self-citations.

As Journal of Clinical Medicine does not strictly forbid salami slicing (“Authors should not unnecessarily divide their work into several related manuscripts”), the continuation of my review will be only based on scientific content.

Overall, the manuscript is well written but Introduction and Discussion parts are too light and require more details.

Line 36 : the traditional technique should be detailed to better understand the difference with RASS

Lines 38-41 : this sentence is not clear, please rephrase

Line 43 : Even if RASS procedure is referenced, it should be defined more consistently in the introduction part

Line 46 : the authors mentioned that retrograde-tunneled HDCs were not available… So the two groups were not compared in the same time ? Are the retrograde-tunneled HDC group the same that the first study ?  (JCM | Free Full-Text | The Rapid Atrial Swirl Sign for Ultrasound-Guided Tip Positioning of Retrograde-Tunneled Hemodialysis Catheters: A Cross-Sectional Study from a Single Center | HTML (mdpi.com)) à salami slicing ?

The introduction part should include more contextual information. The authors should add hypotheses and aim of the study. Some information of the discussion are introductive data. In clinical study, we are not allowed to start without hypothesis and just say “let’s see what would happen”

Parts 2.1 and 2.2 : the time periods are really confusing. It really looks like some patients (retrograde) were included before

Results : overall, the 3 patients that required a replacement of HDCs must be excluded from the analyses because the procedure used for them was different from the other patients. All of analyses must be performed on the 44 remaining patients. So :

Figure 1 : should mention the 3 supplementary exclusions

Line 181 : “decreased body weight” suggest a change over the time. If so, how was it observed ? If not, it should be corrected.

Table 3 : β should be defined in the legend

Lines 192-195 : these sentences are nearly the same as the end of the introduction. Please edit

Lines 199-200 : which advantages ? which studies (details : cross sectional ? single center ? cohort sizes ?) ?

Line 203 : “could be of interest” à hasty conclusion. Should be detailed

Lines 208-209 : table 2 shows normal median platelet counts. Do not confuse low count (normal) and idiopathic thrombocytopenia (pathologic)

Lines 213-216 : this § is not consistent with the manuscript. The RASS procedure is not a strength because it is the only procedure used. The lack of control groups for retrograde- and antegrade-tunneled HDC using traditional procedure is an important limitation of the study. Moreover, the unavailability of retrograde-tunneled HDC is a bias. In fact, the patients of both groups should have been included in the same period and randomly dispatched in the groups. As clinical parameters (table 1) did not differ between the groups, it is not an actual issue, but it should be discussed

References: 12 references for an original article are a very low count, especially when we find 2 self-citations. Nevertheless, edition of introduction and discussion parts would help the authors to add relevant references to support their aim, hypotheses and conclusions.

Author Response

Désirée and Björn Tampe here present a single-center study comparing positioning of retrograde- and antegrade-tunneled hemodialysis catheters (HDC). There are few articles about this topic in literature and this kind of publication usually brings significant knowledges to the community. The procedure they used was previously evaluated for retrograde- (JCM | Free Full-Text | The Rapid Atrial Swirl Sign for Ultrasound-Guided Tip Positioning of Retrograde-Tunneled Hemodialysis Catheters: A Cross-Sectional Study from a Single Center | HTML (mdpi.com)) and then (The rapid atrial swirl sign for ultrasound-guided tip positioning of antegrade-tunneled hemodialysis catheters: A cross-sectional pilot study - Désirée Tampe, Marlene Plüß, Tim Kuczera, Björn Tampe, 2022 (sagepub.com)) for antegrade-tunneled HDC positioning in the same center, by the same authors. Now the authors compare the two types of HDC. They highlighted no differences between the two devices for implantation time. According to their observations, prolonged HDC insertion of antegrade-tunneled HDC was associated with decreased body weight and lower platelet counts.

Thank you for evaluation of our manuscript and valuable comments!

First I have to alert the authors and the editor that this manuscript presents serious breach of scientific integrity. The authors published a study about retrograde-tunneled HDC in September 2021, and a study about antegrade-tunneled HDC in April 2022. Now they submit another study comparing the two devices. The ethics committee approval number was strictly the same. Hence, it really looks like Salami Slicing. Moreover, for only 12 references 2 of them are self-citations.

As Journal of Clinical Medicine does not strictly forbid salami slicing (“Authors should not unnecessarily divide their work into several related manuscripts”), the continuation of my review will be only based on scientific content.

Thank you for pointing that out. We have previously published the feasibility of ultrasound-guided positioning of retrograde hemodialysis catheters, following by application for and antegrade hemodialysis catheters. The aim of these studies was to systematically describe the procedure. We now performed a follow-up study including part of the previously published patients to directly compare ultrasound-guided positioning of retrograde and antegrade hemodialysis catheters, and to identify parameters associated with implantation time. Therefore, we also cited these previous publications in the manuscript. To further clarify this issue, we now include a more detailed description of our previous studies and clearly state the aim of the current study in the revised version of the manuscript. Furthermore, we now include a more detailed introduction and discussion providing additional references on this topic.

Overall, the manuscript is well written but Introduction and Discussion parts are too light and require more details.

Thank you for this positive assessment of our work! As mentioned above, we now include a more detailed introduction and discussion providing additional references on this topic.

Line 36 : the traditional technique should be detailed to better understand the difference with RASS

Thank you for pointing this out! Live fluoroscopic guidance allows for real-time positioning, minor adjustments, and for safe placement of tunneled HDCs. However, this method imposes additional safety risks to the patient and operator due to the radiation. This additional information is now included and referenced in the revised version of the manuscript.

Lines 38-41 : this sentence is not clear, please rephrase

Done accordingly.

Line 43 : Even if RASS procedure is referenced, it should be defined more consistently in the introduction part

Thank you for this suggestion! We now introduced and referenced the RASS in more detail.

Line 46 : the authors mentioned that retrograde-tunneled HDCs were not available… So the two groups were not compared in the same time ? Are the retrograde-tunneled HDC group the same that the first study ?  (JCM | Free Full-Text | The Rapid Atrial Swirl Sign for Ultrasound-Guided Tip Positioning of Retrograde-Tunneled Hemodialysis Catheters: A Cross-Sectional Study from a Single Center | HTML (mdpi.com)) à salami slicing ?

This statement of the referee is correct. As mentioned above, we now include a more detailed description of our previous studies and clearly state the aim of the current study in the revised version of the manuscript.

The introduction part should include more contextual information. The authors should add hypotheses and aim of the study. Some information of the discussion are introductive data. In clinical study, we are not allowed to start without hypothesis and just say “let’s see what would happen”

Thank you for this suggestion! We now provide a more contextual introduction: Previously, we reported that the RASS is an accurate and safe procedure for US-guided tip positioning of retrograde-tunneled HDCs. Due to manufacturer’s reasons, retrograde-tunneled HDCs were not available and antegrade-tunneled HDCs were alternatively implanted at our center. In a separate study, we therefore described feasibility of the RASS for accurate placement of antegrade-tunneled HDCs. In the current study, we aimed to report a comparative analysis of US-guided tip positioning between retrograde- and antegrade-tunneled HDCs, and determinants of implantation time separated for retrograde- and antegrade-tunneled HDCs.

Parts 2.1 and 2.2 : the time periods are really confusing. It really looks like some patients (retrograde) were included before

We apologize for this confusion. The time period was from January 2020 to April 2022. The manuscript was corrected accordingly.

Results : overall, the 3 patients that required a replacement of HDCs must be excluded from the analyses because the procedure used for them was different from the other patients. All of analyses must be performed on the 44 remaining patients. So :

Figure 1 : should mention the 3 supplementary exclusions

We agree with the referee in this important point. We included a total number of 48 patients, one patient was excluded due to failure of HDC insertion. In the remaining 47 patients, 50 HDCs were implanted and three HDCs were excluded because of replacement after the first insertion due to HDC dislocation in the same patients. For the final analysis, 47 HDC insertions were included, separated in 23 retrograde tunneled and 24 antegrade-tunneled HDCs. Figure 1 was modified accordingly.

Line 181 : “decreased body weight” suggest a change over the time. If so, how was it observed ? If not, it should be corrected.

We agree, the term “decreased” was removed. 

Table 3 : β should be defined in the legend

Done as suggested, β is now defined in the corresponding legend.

Lines 192-195 : these sentences are nearly the same as the end of the introduction. Please edit

The part was removed as it was already mentioned in the introduction.

Lines 199-200 : which advantages ? which studies (details : cross sectional ? single center ? cohort sizes ?) ?

Thank you for this comment! We now discuss advantages of the retrograde-tunneled HDC insertion technique, although this has not yet been systematically analyzed. Particularly, the retrograde-tunneled HDC cuff never passes through the exit site, which means the incision can be much smaller pre-venting bleeding and accidental removal of the cuff before it is incorporated. Furthermore, the hub is detachable and enables replacement of broken clamps or hubs without requirement of HDC removal. This information is now included in the revised version of the manuscript.

Line 203 : “could be of interest” à hasty conclusion. Should be detailed

We now discussed and referenced the relevance of HDC implantation time in the revised version of the manuscript, especially since previous observations reported that duration of peripheral and pulmonary arterial catheter placement time was a risk factor for the development of infections, independent of the neutropenic status, or the administration of antibiotics during catheterization. Thank you for pointing this out!

Lines 208-209 : table 2 shows normal median platelet counts. Do not confuse low count (normal) and idiopathic thrombocytopenia (pathologic)

Thank you for pointing this out! We now state that we identified an inverse association between body weight and platelet counts with implantation time specifically in antegrade-tunneled HDCs. In addition, only a minor fraction had thrombocytopenia at time of HDC implantation not differing between the groups. This information is now included in the revised version of the manuscript.

Lines 213-216 : this § is not consistent with the manuscript. The RASS procedure is not a strength because it is the only procedure used. The lack of control groups for retrograde- and antegrade-tunneled HDC using traditional procedure is an important limitation of the study. Moreover, the unavailability of retrograde-tunneled HDC is a bias. In fact, the patients of both groups should have been included in the same period and randomly dispatched in the groups. As clinical parameters (table 1) did not differ between the groups, it is not an actual issue, but it should be discussed

We completely agree with the referee in this important comment! We now discussed strengths and limitations of our study: Our study has several important strengths. First, an important characteristic of our study design was to directly compare tunneled HDC implantation time by applying the RASS for US-guided tip positioning separated for retrograde- and antegrade-tunneled HDCs. Second, a selection bias for either catheter can be excluded because retro-grade-tunneled HDCs were not available due to manufacturer’s reasons. Third, all HDC implantations were performed by a single interventionist, excluding a procedural bias. Finally, relevant clinical and laboratory parameters were comparable in both groups. Our study has also several limitations. First, the relatively small number of patients in a single center and no randomization due to unavailability of retrograde-tunneled HDCs, requiring validation in independent prospective cohorts. Second, the lack of control groups for retrograde- and antegrade-tunneled HDCs using traditional procedure is an important limitation of the study. Third, all tunneled HDCs were inserted through the internal jugular veins and different access sites (e.g., external jugular veins, subclavian veins, and femoral veins) might differ.

References: 12 references for an original article are a very low count, especially when we find 2 self-citations. Nevertheless, edition of introduction and discussion parts would help the authors to add relevant references to support their aim, hypotheses and conclusions.

As suggested, we now added and referenced relevant information in the context of this study in the revised introduction and discussion part of the manuscript. We now performed a follow-up study including part of the previously published patients to directly compare ultrasound-guided positioning of retrograde and antegrade hemodialysis catheters, and to identify parameters associated with implantation time. Therefore, we also cited these previous publications in the manuscript.

Reviewer 2 Report

Dear authors,

Overall, I agree that there may be some differences between anterograde and retrograde permanent catheter (PC) catheter insertion times due to different implantation technique, however I have some key question/remarks.

  1. What anterograde or retrograde catheter was typically inserted in time before the retrograde catheter shortage? I am asking because the doctor's previous preference may have an impact on insertion time.
  2. Slight differences in the time of PC introduction (approximately 15 min) from the clinical point of view are of little value. It would be more important to evaluate any urgent complication or to analyze the position of the tip.
  3. I get the impression, that the study is underpowered, because in multiple linear regression analysis the minimum sample size for the 9 variables in Figure 3 should include a minimum of 122 procedures according to the n≥50 + 8xk rule (Samuel B. Green, How Many Subjects Does It Take To Do A Regression Analysis. Multivariate Behavioral Research, https://doi.org/10.1207/s15327906mbr2603_7) and we have only 47!
  4. Was fluoroscopy used to check guidewire position? It is necessary to carefully describe the procedure for inserting a retrograde and retrograde catheter.
  5. The two main conclusions of this study are that prolonged PC insertion times are associated with decreased body weight (BW) and lower platelet counts. I can only imagine that a lower platelet count would be important information for the operator to carry out the procedure more carefully and longer. On the other hand, it is difficult to explain why thin patients are more difficult than obese patients and take longer to operate - logically, the relationship should be the opposite!

Minor remarks

Table 1. Etiology of ESRD - For a small number of diseases, I suggest creating more general groups of causes, for example, "Glomerulonephritis"

RASS (Rapid Atrial Swirl Sign) are used to locate the catheter tip, but from my point of view in anterograde PC insertion, this method has only limited relevance as the tunneling of the catheter precedes the insertion of the catheter through the peel away sheet. This makes it difficult to relocate the catheter because it risks advancing the cuff beyond the exit side.

Author Response

Dear authors,

Overall, I agree that there may be some differences between anterograde and retrograde permanent catheter (PC) catheter insertion times due to different implantation technique, however I have some key question/remarks.

We thank the referee for her/his positive evaluation of our manuscript. Please find a point-to-point response to your comments below.

  1. What anterograde or retrograde catheter was typically inserted in time before the retrograde catheter shortage? I am asking because the doctor's previous preference may have an impact on insertion time.

Previously, we have implanted retrograde-tunneled HDCs due to advantages that are also discussed in the manuscript. We now include a statement that retrograde-tunneled HDCs are preferred in our center.

  1. Slight differences in the time of PC introduction (approximately 15 min) from the clinical point of view are of little value. It would be more important to evaluate any urgent complication or to analyze the position of the tip.

We agree with the referee in this comment. We did not observe differences in procedural complications or HDC tip positions. However, our observation that antegrade-tunneled HDC implantation tend to be faster might be of relevance, and identification of patient subgroups who might benefit from especially antegrade-tunneled HDCs requires further investigation. This is especially relevant since previous observations reported that duration of peripheral and pulmonary arterial catheter placement time was a risk factor for the development of infections, independent of the neutropenic status, or the administration of antibiotics during catheterization. This information is now included and referenced in the revised version of the manuscript.

  1. I get the impression, that the study is underpowered, because in multiple linear regression analysis the minimum sample size for the 9 variables in Figure 3 should include a minimum of 122 procedures according to the n≥50 + 8xk rule (Samuel B. Green, How Many Subjects Does It Take To Do A Regression Analysis. Multivariate Behavioral Research, https://doi.org/10.1207/s15327906mbr2603_7) and we have only 47!

Thank you for this important comment! To address this issue, we performed univariate Spearman's correlations as shown in Figure 3. A Spearman's ρ more than ±0.4 in the correlation matrix was defined as relevant and indicated by rectangle boxes in the corresponding figure. For multiple linear regression, an independent statistical evaluation of these parameters was performed as shown in Table 3. This information is now clearly stated in the method section and corresponding legends of Figure 3 and Table 3.

  1. Was fluoroscopy used to check guidewire position? It is necessary to carefully describe the procedure for inserting a retrograde and retrograde catheter.

We apologize for this missing information! We now provide a more contextual introduction: Live fluoroscopic guidance allows for real-time positioning, minor adjustments, and for safe placement of tunneled HDCs. However, this method imposes additional safety risks to the patient and operator due to the radiation. It has already been shown that the conversion from a non-tunneled to a tunneled HDC without using fluoroscopy in an incident dialyzed cohort is safe, but also inexpensive [6]. For HDC placement, the agitated bubble-enhanced visualization has been used as a mixture of 9 mL of normal saline solution and 1 mL of air and has been shown to be a safe procedure for tunneled HDC insertion without fluoroscopy. However, rare events of ischemic cerebrovascular complications in patients with cardiac or intrapulmonary shunts have been reported and attributed to air bubbles. Alternatively, the rapid atrial swirl sign (RASS) has also been described for placement of central venous catheters. In this procedure, 10 mL of normal saline is flushed into the distal HDC port. The RASS is defined as the ultrasound (US)-visualized appearance of turbulences entering the right atrium immediately after the saline flush and migrating towards the right ventricle. This information is now included in the revised version of the manuscript.

  1. The two main conclusions of this study are that prolonged PC insertion times are associated with decreased body weight (BW) and lower platelet counts. I can only imagine that a lower platelet count would be important information for the operator to carry out the procedure more carefully and longer. On the other hand, it is difficult to explain why thin patients are more difficult than obese patients and take longer to operate - logically, the relationship should be the opposite!

Thank you for pointing this out! We now state that we identified an inverse association between body weight and platelet counts with implantation time specifically in antegrade-tunneled HDCs. In addition, only a minor fraction had thrombocytopenia at time of HDC implantation not differing between the groups. Also, we did not observe a correlation with the body mass index (BMI) itself, and longer implantation time was associated with less body weight. This information is now included in the revised version of the manuscript.

Minor remarks

Table 1. Etiology of ESRD - For a small number of diseases, I suggest creating more general groups of causes, for example, "Glomerulonephritis"

Done accordingly.

RASS (Rapid Atrial Swirl Sign) are used to locate the catheter tip, but from my point of view in anterograde PC insertion, this method has only limited relevance as the tunneling of the catheter precedes the insertion of the catheter through the peel away sheet. This makes it difficult to relocate the catheter because it risks advancing the cuff beyond the exit side.

For the placement of the antegrade-tunneled HDC, a guide wire was inserted for venous dilation and the HDC was inserted through the peel-apart introducer sheath. The RASS was used for tip positioning and the exit site was defined. Finally, the HDC was tunneled antegrade under local anesthesia and inserted after peeling away the introducer sheath. Therefore, the RASS was also used for antegrade-tunneled HDC tip positioning before definition of the exit site and tunnel preparation. This information is now included in the revised version of the manuscript.

Round 2

Reviewer 1 Report

Dear authors,

The manuscript was revised according to the comments of the reviewers.

Author Response

Dear authors,

The manuscript was revised according to the comments of the reviewers.

We thank the referee again for her/his time to evaluate our study. Due to the referee's important comments, we also belief that the manuscript has significantly improved. 

Reviewer 2 Report

Low number procedures probably led to unexpected results and the implantation time was related to the patient's weight and thrombocytopenia. The slight reduction in the time of anterograde catheter implantation has low or no clinical significance.

Author Response

Low number procedures probably led to unexpected results and the implantation time was related to the patient's weight and thrombocytopenia. The slight reduction in the time of anterograde catheter implantation has low or no clinical significance.

We thank the referee again for these comments! We agree that the relatively small number of patients in a single center and no randomization is a limitation of this study. This is clearly stated in the revised version of the manuscript and requires independent validation. Regarding the reduced implantation time of antegrade-tunneled HDCs, previous observations reported that duration of peripheral and pulmonary arterial catheter placement time was a risk factor for the development of infections, independent of the neutropenic status, or the administration of antibiotics during catheterization. However, we agree that this observation is only preliminary and need further validation with regard to risk for the development of infectious complications. This issue is now discussed in the revised version of the manuscript.